# Equipping LLMs with Self-Awareness for High-Stakes Tasks

## Abstract

Overconfidence in large language model responses has emerged as a critical barrier for deploying these systems in high-stakes tasks such as cyber threat intelligence, financial analysis, and clinical decision support. This issue stems from reward-optimal behavior, as LLMs are trained to produce answers even under uncertainty. Nevertheless, most approaches in high-stakes domains continue to treat these tasks as primarily knowledge-intensive, focusing on scaling, retrieval, or fine-tuning, while leaving the problem of overconfidence unresolved. Recent studies have begun to highlight this gap, calling for solutions that move beyond superficial calibration or knowledge expansion. Building on these challenges, we identify self-awareness as a missing capability for LLMs in high-stakes deployment: the ability to recognize the limits of *what they know* and to assess the certainty of *how well they know it*. To this end, we propose a framework that trains LLMs to cultivate self-awareness via reinforcement learning and decouples awareness learning from task performance, pairing it with adaptive inference-time strategies such as retrieval-augmented generation and low-confidence regeneration. We evaluate our framework in the cybersecurity domain. Results demonstrate that our method substantially reduces confidently wrong outputs, surpassing the strongest baseline by up to 95.6%, while achieving competitive performance. Our implementations are available at https://anonymous.4open.science/r/SelfAwareLLM.

## 1 Introduction

Large language models (LLMs) have achieved impressive success on knowledge-intensive tasks (Kamalloo et al., 2023; Wang et al., 2024) such as open-domain question answering and scientific analysis, underscoring their potential as powerful knowledge systems. Yet many real-world applications are not only knowledge-intensive but also high-stakes (Alam et al., 2024; Koa et al., 2024), meaning that errors can carry severe consequences. For instance, misleading an analyst about an emerging cyber threat, misjudging market volatility, or misguiding a clinical decision may incur significant risks to security, financial stability, or human well-being. In these settings, success requires more than being knowledgeable: models also need to be faithful, meaning they can provide responses that are not only informative but also aware of *what they know* and *how well they know it*, thus preserving reliability under uncertainty.

Nevertheless, most existing approaches to high-stakes tasks fall back on a familiar recipe: treating them as knowledge-intensive and framing the challenges as knowledge deficits (Wei et al., 2025b). Typical strategies include scaling model size, retrieval-augmented generation (RAG), or fine-tuning on domain-specific corpora (Lewis et al., 2020; Kaplan et al., 2020). While these methods often improve accuracy and even yield seemingly better calibration scores, they leave reward-optimal overconfidence unresolved (Kalai et al., 2025; Mezzi et al., 2025). As a result, the apparent improvements in reliability do not translate into genuine faithfulness.

In parallel, efforts to improve faithfulness have largely focused on confidence calibration (Chhikara, 2025). Yet aligning expressed confidence with actual correctness has proven notoriously difficult. Current training paradigms often induce reward-optimal overconfidence (Kalai et al., 2025): models are optimized to provide answers whenever possible, even under uncertainty, and frequently express high confidence in factually incorrect outputs. This phenomenon is illustrated in Figure 1(b), which

Figure 1: Illustration of the two components of self-awareness. (a) **Boundary-awareness**: within a domain-specific knowledge space, the model only possesses a subset through pretraining; it must distinguish between IB and OOB queries. (b) **Confidence-awareness**: models often assign similarly high confidence to both correct and incorrect answers; confidence-awareness seeks to reduce unwarranted confidence in wrong outputs.

shows that the model assigns similarly high confidence to both correct and incorrect answers. Crucially, this phenomenon cannot be reduced to ordinary hallucination alone; it stems from training objectives that prioritize fluency and coverage over calibrated reliability (Turpin et al., 2023). In high-stakes contexts, such confidently wrong outputs are intolerable, as faithfulness is just as essential as correctness itself.

These challenges call for a paradigm shift for high-stakes tasks: faithfulness must be treated as a first-class objective rather than a by-product of knowledge expansion. Scaling models and augmenting retrieval will undoubtedly enhance what LLMs can access, but knowledge alone is insufficient. What makes high-stakes scenarios particularly demanding is not just the need for accurate answers, but the cost of confidently wrong ones: errors expressed with unwarranted certainty can mislead decision-makers and compromise model safety. This highlights the central role of **self-awareness**, which we conceptualize as comprising two complementary components. First, *boundary-awareness* is the ability to recognize the limits of what a model knows. As illustrated in Figure 1(a), within a domain-specific knowledge space the model possesses only a subset, and boundary-awareness enables distinguishing the known region from the unknown. Second, *confidence-awareness* is the ability to assess how certain the model is about its own responses, that is, to gauge the degree of confidence it can justifiably assign to an answer within the boundary.

Building on the above framing, we equip LLMs with self-awareness rather than merely expanding knowledge or rewarding correctness. Concretely, we design reinforcement learning objectives that explicitly cultivate two complementary capabilities: *boundary-awareness* and *confidence-awareness*. By focusing on awareness signals rather than correctness, the model learns to avoid confidently wrong outputs even when it cannot provide the right answer. Based on this learned self-awareness, we develop a set of adaptive inference-time strategies, including retrieval-augmented generation and low-confidence regeneration, that regulate response generation. A key feature of our design is that awareness learning and adaptive strategies are decoupled: awareness is trained independently and remains a "pure" signal about knowledge and certainty, unshaped by task performance incentives. This separation ensures that models can reliably use self-awareness to guide behavior under uncertainty, leading to more faithful decision-making in high-stakes tasks.

To validate the effectiveness of our framework, we conduct extensive experiments on diverse cybersecurity benchmarks. We focus on this domain because it is both high-stakes, where errors can have substantial consequences, and data-scarce, making large-scale supervised learning difficult. Our results show that an LLM equipped with self-awareness alone significantly reduces confidently wrong outputs, outperforming the strongest baseline by up to 95.6%, while maintaining competitive task performance. When combined with adaptive inference-time strategies, the confidently wrong rate remains low, and task performance is boosted. These findings validate that treating faithfulness as a first-class objective enables LLMs to be equipped with self-awareness, providing a practical path toward safer deployment in high-stakes domains.

## 2 RELATED WORK

**LLMs for High-Stakes Tasks** LLMs are increasingly applied in high-stakes domains such as cybersecurity, law, finance, and healthcare, where adaptation typically relies on three paradigms:

in-context learning, fine-tuning, and domain-specific pretraining. **In-context learning** adapts models without parameter updates through prompt design, retrieval, or reasoning scaffolds; for example, retrieval-augmented generation has been applied to law (Li et al., 2025a), finance (Zhao et al., 2024), and healthcare (Zhao et al., 2025a), while Chain-of-Thought prompting and o1-type models improve reasoning via step-by-step prompting with multi-pass search (Wei et al., 2025a). **Fine-tuning** provides a cost-effective adaptation strategy: Chu et al. (2025) propose Domain$o$1s, a reasoning model for high-stakes domains, while other efforts adapt models for legal reasoning (Guha et al., 2023), clinical decision support (Rajashekar et al., 2024), and phishing detection (Trad & Chehab, 2024). Lightweight techniques such as synthetic data generation (Wei et al., 2025b; Hsu et al., 2024) further reduce costs, and benchmarks like PIXIU (Xie et al., 2023) showcase benefits in finance. **Domain-specific pretraining** captures specialized terminology and reasoning patterns at scale, exemplified by BloombergGPT for finance (Wu et al., 2023), though such approaches demand prohibitive data and compute resources (Shi et al., 2025; Chu et al., 2025; Xie et al., 2023). While these strategies improve accuracy and domain adaptation, they largely treat challenges as knowledge deficits, leaving reward-optimal overconfidence unresolved.

**Knowledge Boundary**  LLMs combine parametric knowledge stored in model weights with external knowledge accessible via retrievers (Zheng et al., 2024). The concept of a knowledge boundary delineates the limits of internal knowledge (Li et al., 2024; Xu et al., 2024a), typically probed either by template-based prompts for factual recall (Petroni et al., 2019) or by internal-state analyses (Chen et al., 2024; Zhao et al., 2025b; Huang et al., 2025). Understanding these boundaries is particularly important for retrieval-augmented generation (RAG), enabling models to decide when to rely on external sources and thereby mitigate hallucinations (Ren et al., 2025; Huang et al., 2025).

**Confidence Estimation and Calibration**  LLMs are known to exhibit systematic overconfidence, often assigning high probabilities or verbal certainty to incorrect outputs (Turpin et al., 2023; Kalai et al., 2025; Leng et al., 2025). This is especially problematic in high-stakes contexts, where confidently wrong answers can mislead decision-makers more severely than simple uncertainty. To mitigate this, researchers have proposed confidence estimation and calibration methods. Estimation methods are either **white-box**, probing hidden states to predict correctness or truthfulness (Kadavath et al., 2022; Azaria & Mitchell, 2023; Orgad et al., 2024), or **black-box**, relying on outputs via linguistic markers (Xiong et al., 2024; Lin et al., 2022), sampling-based self-consistency (Manakul et al., 2023; Kuhn et al., 2023), or surrogate models (Shrivastava et al., 2023). Building on these, calibration methods seek to align expressed confidence with correctness through token- and sequence-level calibration (Zhao et al., 2022), linguistic calibration (Lin et al., 2022; Zhou et al., 2023), or reinforcement learning for optimizing verbalized confidence (Xu et al., 2024b; Damani et al., 2025). Yet despite metric improvements, calibration rarely reduces confidently wrong cases in practice, as faithfulness is still treated as a by-product of accuracy optimization rather than a first-class training objective. Our work instead seeks to directly optimize self-awareness to mitigate reward-optimal overconfidence.

## 3 METHODOLOGY

In this section, we present our methodology for operationalizing self-awareness. We first estimate the model's knowledge boundary (Section 3.1), and then apply reinforcement learning to internalize both boundary awareness and confidence awareness (Section 3.2). Building on the established self-awareness, we develop adaptive test-time strategies that improve both the faithfulness and accuracy of model generation (Section 3.3).

### 3.1 ESTIMATION OF KNOWLEDGE BOUNDARY

Here, the term *knowledge boundary* characterizes the separation between knowledge that can be accessed from a model's inner-parameters (*in-boundary, IB*) and knowledge that lies outside of it (*out-of-boundary, OOB*) (Huang et al., 2025; Li et al., 2024). In practice, this notion can also be applied at the query level: a query is considered IB if the knowledge required to answer it resides within the model's parametric capacity, and OOB otherwise. Since the boundary is not explicitly defined within the model, we follow prior work in estimating it as an external supervisory signal (Huang et al., 2025). To approximate this boundary, we probe the model with a set of queries $\mathcal{Q}$.

Figure 3: A toy example of the reinforcement learning process with designed rewards.

Specifically, for each query $q \in \mathcal{Q}$, we sample $N$ responses (answers) with in-context learning. If the correct answer appears *at least once*, $q$ is labeled as in-boundary ($q \in \mathcal{Q}_{\text{IB}}$), indicating the model likely possesses the relevant knowledge; otherwise, it is labeled as out-of-boundary ($q \in \mathcal{Q}_{\text{OOB}}$). The resulting partition $\mathcal{Q} = \mathcal{Q}\text{IB} \cup \mathcal{Q}\text{OOB}$ (illustrated in Figure 2) provides a learnable boundary signal that separates IB from OOB queries.

## 3.2 JOINT REINFORCEMENT LEARNING FOR SELF-AWARENESS

To equip the model with self-awareness, encompassing both *what it knows* (boundary-awareness) and *how well it knows* (confidence-awareness), we frame awareness as the bridge from being merely knowledgeable to becoming faithful. Rather than a static property, self-awareness here is defined as an optimizable objective: boundary cues and self-reported confidence are formalized into reward signals that reinforcement learning can refine.

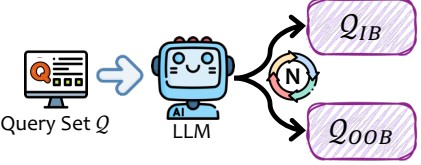

Figure 2: An illustration of estimating knowledge boundary through queries.

**Boundary-Aware Reward.** Building on the estimated knowledge boundary (Section 3.1), we define a reward that encourages the model to distinguish between IB and OOB queries. For a query $q$, let $z \in \{0, 1\}$ denote its OOB status ($z = 1$ if $q \in \mathcal{Q}_{\text{OOB}}$, $z = 0$ otherwise). The boundary-aware reward is then defined as:

$$R_{\text{boundary}}(q, z) = \mathbb{1}\{f_\theta(q) \equiv z\}, \tag{1}$$

where $f_\theta(q)$ is the model's predicted OOB status, obtained through an explicit self-reported boundary prediction. Note that, this objective does not expand the model's underlying knowledge; instead, it encourages explicit recognition of knowledge limits, consolidating boundary awareness as a core component of self-awareness.

**Confidence-Aware Reward.** Beyond boundary recognition, an equally important dimension of self-awareness is to evaluate how well an answer is supported. For this purpose, we use the model's self-reported confidence, elicited by prompts that request a verbal confidence rating alongside the generated answer. Such ratings provide a soft and optimizable signal, unlike logit-based uncertainty (Kang et al., 2025; Zhang et al., 2025), which is difficult to directly refine, enabling reinforcement learning to align expressed confidence with actual correctness. Formally, the confidence-aware reward is defined as:

$$R_{\text{conf}}(q, a, a^*) = -\Big( \text{conf}(q, a) - \mathbb{1}\{a \equiv a^*\} \Big)^2, \tag{2}$$

which corresponds to a Brier-style score (Glenn et al., 1950), penalizing the squared deviation between reported confidence and correctness. This design provides a soft, optimizable signal, rather than a rigid calibration objective, thereby consolidating confidence awareness as a core component of self-awareness.

**Joint Optimization.** The overall reward integrates boundary awareness, confidence awareness, and the standard practice of format validity (Guo et al., 2025). Formally, it is defined as:

$$R = \begin{cases} 0 & \text{if the output format is invalid,} \\ R_{\text{boundary}} + R_{\text{conf}} & \text{otherwise.} \end{cases} \tag{3}$$

We adopt Group Relative Policy Optimization (GRPO, Shao et al. (2024)) as the reinforcement learning algorithm, which stabilizes training by normalizing rewards across groups of rollouts.

Given a reference policy $\pi_{\text{ref}}$ and an old policy $\pi_{\theta_{\text{old}}}$, we sample $G$ rollouts $y_i \sim \pi_{\theta_{\text{old}}}(\cdot \mid x)$ for each input $x \sim \mathcal{D}$, and update $\pi_\theta$ by maximizing:

$$
\begin{aligned}
J_{\text{GRPO}}(\theta) = \mathbb{E}_{x \sim \mathcal{D},\, y_i \sim \pi_{\theta_{\text{old}}}} \Bigg[ & \frac{1}{G} \sum_{i=1}^{G} \min\Big( \frac{\pi_\theta(y_i \mid x)}{\pi_{\theta_{\text{old}}}(y_i \mid x)} A_i, \\
& \text{clip}\Big( \frac{\pi_\theta(y_i \mid x)}{\pi_{\theta_{\text{old}}}(y_i \mid x)}, 1 - \varepsilon,\, 1 + \varepsilon \Big) A_i \Big) - \beta\, D_{\text{KL}}\big(\pi_\theta \,\|\, \pi_{\text{ref}}\big) \Bigg],
\end{aligned}
\tag{4}
$$

where $A_i$ is the normalized advantage of the $i$-th rollout. The reference policy $\pi_{\text{ref}}$ is an instruction-tuned baseline, while $\varepsilon$ and $\beta$ control clipping and KL penalty strength (Schulman et al., 2017).

**Curriculum-Style Training Strategy.** Jointly optimizing all reward components from the outset can lead to unstable learning, as multiple objectives introduce noisy and conflicting signals—an issue that becomes more pronounced under limited knowledge-intensive data (Bengio et al., 2009; Liu et al., 2025). To address this, we adopt a curriculum-style strategy that gradually expands the optimization scope. In the first phase, training focuses only on format and confidence-aware rewards, which are easier to learn since they rely on self-reported signals and require no additional supervision. Once this foundation is established, the boundary-aware reward is introduced in the second phase, enabling the model to progressively acquire awareness of its knowledge limits. This staged optimization reduces variance, improves sample efficiency, and stabilizes training.

### 3.3 ADAPTIVE STRATEGIES WITH SELF-AWARENESS

Building on the established self-awareness, we move from recognizing limitations to leveraging this awareness for improving reliability in knowledge-intensive tasks. While the preceding components focus on faithfulness in high-stakes scenarios, self-awareness alone does not guarantee effective task

Table 1: Adaptive strategies derived from boundary and confidence awareness.

| Boundary | Confidence | Strategy |
|----------|-----------|----------|
| In | High | Apply the current answer |
| In | Low | Re-query until high confidence |
| Out | - | Apply RAG to generate answer |

performance. To address this, boundary and confidence signals are operationalized into adaptive strategies that guide response generation, ensuring that the model not only knows its limits but also acts upon them to deliver more dependable answers. Unlike prior approaches that couple such strategies directly into the reinforcement learning objective (Huang et al., 2025; Jin et al., 2025; Wei et al., 2025c), our design explicitly decouples learning and adaptation: the reinforcement learning stage focuses purely on optimizing awareness signals, while adaptive strategies independently act on these signals during inference. This separation allows each component to specialize in its role, providing clearer training objectives and greater flexibility at deployment. The resulting strategies are summarized in Table 1.

## 4 EXPERIMENTS

### 4.1 SETUPS

**Dataset** We conduct our evaluation in the domain of cybersecurity, a representative high-stakes, knowledge-intensive setting. We evaluate on six datasets: four in-distribution (ID) benchmarks used during reinforcement learning and two held-out out-of-distribution (OOD) datasets. The in-distribution benchmarks are CSEBench (Wang et al., 2025), CyberMetric (Tihanyi et al., 2024), SEvenLLM (Ji et al., 2024), and the AttackSeqBench_Tactic subset (Yong et al., 2025). For out-of-distribution evaluation, we adopt CTIBench (Alam et al., 2024) and the AttackSeqBench_Technique subset (Yong et al., 2025). Out-of-Distribution evaluation allows us to assess generalization under distribution shift. Following the estimation procedure in Section 3.1, each dataset is further partitioned into In-Boundary and Out-of-Boundary subsets. We follow common practice by splitting each dataset into training/validation/test sets with a 75/10/15 ratio. To prevent shortcut learning and ensure fair exposure, we apply balanced sampling in training, maintaining a 1:1 ratio between IB and OOB queries. Detailed dataset statistics are provided in the Appendix B.

**Baselines** We compare our method against two categories of baselines. The first group includes general-purpose models and training approaches: (i) Qwen2-7B-Instruct (Team, 2024), (ii) a Naive RAG variant with non-selective retrieval-augmented generation, (iii) RLVR (Guo et al., 2025), which applies reinforcement learning with verifiable rewards based on answer correctness, and (iv) RLCR (Damani et al., 2025), which adopts a calibration reward to align expressed confidence with correctness. The second group consists of domain-specific security LLMs fine-tuned on cybersecurity corpora, including Primius-Reasoning, Primius-Merged (Yu et al., 2025), and SecurityLLM[1].

**Evaluation Metrics** We evaluate model performance from three perspectives:

- **Task Performance.** We report **Accuracy** ($\uparrow$) as the measure of task performance.

- **Confidence Calibration.** To evaluate the alignment between the model's confidence and correctness, we report the **Brier** Score ($\downarrow$) (Glenn et al., 1950), Expected Calibration Error (**ECE**; $\downarrow$) (Guo et al., 2017), and the Area Under the ROC Curve (**AUROC**; $\uparrow$) (Hanley & McNeil, 1982). The calculation of these metrics is given in the Appendix B.

- **Risk of Confident Errors.** We propose **Confident Wrong Rate** (**CWR@$\tau$**) to quantify the proportion of errors made with confidence above a threshold $\tau$. Unlike standard accuracy-based metrics that treat all errors equally, CWR@$\tau$ focuses specifically on high-confidence mistakes, thereby capturing the risk of being confidently wrong. Lower values indicate a reduced tendency to produce such errors. Formally, let $E$ denote the set of incorrect predictions,

$$\text{CWR@}\tau = \frac{1}{|E|} \sum_{a \in E} \mathbb{1}\{\text{conf}(q, a) > \tau\}. \tag{5}$$

**Implementation Details** We conduct all experiments using 8 NVIDIA A800-80G GPUs for training and 2 NVIDIA A100-80G GPUs for evaluation. Our implementation is based on Qwen2-7B-Instruct, trained with the GRPO reinforcement learning algorithm within the `verl` framework (Sheng et al., 2025). For retrieval, we adopt ATTACK-BERT[2] (Abdeen et al., 2023) as the embedding model, FAISS (Douze et al., 2024) for similarity search, and the MITRE ATT&CK corpus[3] as the external knowledge base. Each tactic, technique, or procedure (TTP) entry is treated as an individual retrieval unit. Key hyperparameters are listed in Table 2.

Table 2: Parameter settings.

| Parameter | Value |
|---|---|
| Times of Sample for Estimation | 16 |
| Learning Rate | 1e-6 |
| Train Batch Size | 256 |
| Number of Training Epochs | 3 |
| Number of Rollout | 5 |
| Rollout Temperature | 1.0 |
| KL Loss Coefficient | 0.001 |
| Clip Ratio | 0.2 |

### 4.2 RESULTS

#### 4.2.1 MAIN RESULTS

We analyze the main results from three perspectives: (i) comparisons with general methods, to examine how different optimization objectives affect confidence; (ii) comparisons with domain-specific LLMs, to assess the role of injected knowledge; and (iii) task performance and the generalization to OOD datasets. Tables 3 and 4 summarize the results on ID and OOD benchmarks. Overall, *our method substantially reduces confidently wrong outputs*: while most baselines exhibit CWR@0.8 values close to 100%, our framework achieves a marked reduction, particularly relative to its base model Qwen2.5-7B-Instruct.

**Comparison with General Methods.** Among general approaches, RLCR is designed to improve calibration. As the results show, it achieves modest reductions in CWR on the in-distribution SEvenLLM dataset but performs poorly on OOD benchmarks, likely because its optimization jointly pursues calibration and accuracy, diluting the effect on awareness. In contrast, RLVR demonstrates strong task accuracy, which is expected since correctness is its sole reward signal. However, its

---

[1] https://huggingface.co/ZySec-AI/SecurityLLM
[2] https://huggingface.co/basel/ATTACK-BERT
[3] https://attack.mitre.org/

Table 3: In-distribution evaluation results. The best scores are shown in **bold**, and the second-best in *italic underline*. "Our (LLM)" denotes the model trained with self-awareness only, while "Our (Framework)" includes both self-awareness and adaptive strategies.

| Benchmarks | CSEBench | | | | | CyberMetric | | | | |
|---|---|---|---|---|---|---|---|---|---|---|
| Metrics | Acc. (%)↑ | Brier ↓ | ECE ↓ | AUROC ↑ | CWR@0.8 (%)↓ | Acc. (%)↑ | Brier ↓ | ECE ↓ | AUROC ↑ | CWR@0.8 (%)↓ |
| Qwen2.5-7B-Instruct | 42.8 | 0.46 | 0.47 | *0.64* | 100 | 43.9 | 0.45 | 0.47 | *0.64* | 100 |
| RLVR | 47.6 | *0.44* | *0.44* | 0.57 | 100 | 49.5 | *0.42* | *0.42* | 0.60 | 100 |
| RLCR | 47.2 | 0.50 | 0.50 | 0.53 | 100 | 47.2 | 0.49 | 0.49 | 0.57 | 100 |
| Base + Naïve RAG | 40.8 | 0.48 | 0.49 | 0.56 | 98.0 | 46.2 | 0.43 | 0.43 | 0.55 | 95.6 |
| SecurityLLM | 00.4 | 0.92 | 0.96 | **0.72** | 100 | 00.5 | 0.94 | 0.97 | **0.69** | 100 |
| Primus-Merged | **53.6** | **0.35** | **0.34** | 0.57 | **94.8** | **57.1** | **0.34** | **0.31** | 0.55 | 98.9 |
| Primus-Reasoning | *50.8* | 0.49 | 0.49 | 0.50 | 100 | *54.3* | 0.46 | 0.46 | 0.50 | 100 |
| Ours (LLM) | 44.0 | 0.55 | 0.55 | 0.52 | *95.0* | 47.6 | 0.48 | 0.48 | 0.52 | **86.0** |
| Ours (Framework) | 44.4 | 0.55 | 0.56 | 0.50 | 95.7 | 48.1 | 0.50 | 0.50 | 0.50 | *95.5* |

| Benchmarks | SEvenLLM | | | | | AttackSeqBench_Tactic | | | | |
|---|---|---|---|---|---|---|---|---|---|---|
| Metrics | Acc. (%)↑ | Brier ↓ | ECE ↓ | AUROC ↑ | CWR@0.8 (%)↓ | Acc. (%)↑ | Brier ↓ | ECE ↓ | AUROC ↑ | CWR@0.8 (%)↓ |
| Qwen2.5-7B-Instruct | 40.0 | 0.46 | 0.47 | 0.53 | 99.3 | 43.9 | 0.46 | 0.46 | 0.56 | 100 |
| RLVR | **55.8** | **0.34** | **0.31** | 0.51 | 100 | **61.5** | **0.31** | **0.28** | 0.62 | 100 |
| RLCR | *45.4* | 0.47 | 0.46 | 0.51 | 92.4 | *56.1* | 0.42 | 0.41 | 0.54 | 100 |
| Base + Naïve RAG | 38.3 | 0.48 | 0.49 | 0.49 | 91.9 | 33.8 | 0.55 | 0.57 | 0.51 | 100 |
| SecurityLLM | 00.8 | 0.91 | 0.94 | **0.72** | 99.2 | 00.0 | 0.92 | 0.96 | 0.50 | 100 |
| Primus-Merged | 43.3 | *0.41* | *0.42* | *0.65* | 92.7 | 49.3 | 0.39 | *0.38* | 0.55 | 96.0 |
| Primus-Reasoning | 44.6 | 0.54 | 0.54 | 0.51 | 97.7 | 38.5 | 0.61 | 0.61 | 0.50 | 100 |
| Ours (LLM) | 41.3 | 0.47 | 0.47 | 0.54 | **61.7** | 48.7 | 0.41 | 0.42 | *0.61* | **55.3** |
| Ours (Framework) | 39.6 | 0.49 | 0.51 | 0.59 | *77.2* | 54.7 | *0.38* | 0.38 | **0.62** | *68.7* |

CWR values remain close to 100%, reflecting the typical overconfidence of reward schemes that optimize only for correctness while neglecting faithfulness. Interestingly, the base model augmented with naïve RAG does not exhibit such extreme CWR values. We hypothesize this arises from contradictions between retrieved knowledge and parametric knowledge, or from retrieval failures when the external corpus lacks the necessary information. In such cases, irrelevant or conflicting evidence may disrupt the model's confidence, unintentionally exposing its uncertainty, though this also explains why its task performance is often worse than the base model. This observation highlights the *importance of detecting knowledge boundaries*, especially in data-scarce domains where retrieval is prone to noise and cannot reliably provide the required knowledge.

**Comparison with Domain-Specific LLMs**  Domain-specific LLMs exhibit more mixed behaviors. SecurityLLM shows extremely low accuracy, often producing verbose, knowledge-like text without addressing the query itself. Prior analyses (Fu et al., 2025; Li et al., 2025b) suggest that fine-tuning on domain corpora may have corrupted its instruction, following ability, which explains its anomalously high AUROC values, scores that are misleading rather than meaningful. In contrast, the Primus series occasionally achieves lower CWR values (e.g., on CSEBench), but overall remains inconsistent across benchmarks. These observations indicate that while domain-specific knowledge can improve performance in isolated cases, it does not reliably mitigate overconfidence. This highlights the potential of combining self-awareness with domain-targeted knowledge injection as a more promising future direction.

**Task Performance and Generalization to OOD**  From the perspective of task performance, we observe that most models, except RLVR and the reasoning variants of Primus, show relatively small differences in accuracy. Against this backdrop, our framework consistently outperforms the base model and is often comparable to RLCR, though it lags behind RLVR on some benchmarks. Given that our optimization objective does not directly target correctness, this gap is expected and acceptable. Moreover, our adaptive inference-time strategies provide additional performance gains on top of the trained self-aware model, confirming their utility. However, these improvements remain limited, likely due to the restricted scale of our knowledge base, which prevents effective retrieval in many cases. Beyond accuracy, self-awareness also improves robustness under distribu-

Table 4: Results of out-of-distribution evaluation. The best scores are shown in **bold**, and the second-best in *italic underline*. "Our (LLM)" denotes the model trained with self-awareness only, while "Our (Framework)" includes both self-awareness and adaptive strategies.

| Benchmarks | CTIBench | | | | | AttackSeqBench_Technique | | | | |
|---|---|---|---|---|---|---|---|---|---|---|
| Metrics | Acc. (%)↑ | Brier ↓ | ECE ↓ | AUROC ↑ | CWR@0.8 (%)↓ | Acc. (%)↑ | Brier ↓ | ECE ↓ | AUROC ↑ | CWR@0.8 (%)↓ |
| Qwen2.5-7B-Instruct | 34.9 | 0.51 | 0.54 | *0.62* | 97.1 | 37.5 | 0.49 | 0.52 | **0.68** | 100 |
| RLVR | 40.6 | *0.47* | *0.48* | 0.55 | 100 | 38.8 | 0.48 | 0.50 | *0.62* | 100 |
| RLCR | 34.9 | 0.58 | 0.60 | 0.60 | 100 | 36.3 | 0.58 | 0.60 | 0.58 | 100 |
| Base + Naïve RAG | *40.7* | 0.47 | 0.48 | 0.57 | 95.2 | 43.8 | *0.45* | **0.46** | 0.61 | 97.8 |
| SecurityLLM | 00.9 | 0.91 | 0.95 | **0.77** | 100 | 00.0 | 0.90 | 0.95 | 0.50 | 100 |
| Primus-Merged | **50.0** | **0.39** | **0.37** | 0.50 | 98.1 | *47.5* | **0.41** | 0.39 | 0.47 | 97.6 |
| Primus-Reasoning | 38.7 | 0.61 | 0.61 | 0.50 | 100 | **51.3** | 0.49 | *0.49* | 0.50 | 100 |
| Ours (LLM) | 32.1 | 0.59 | 0.61 | 0.58 | **72.2** | 32.5 | 0.50 | 0.53 | 0.60 | **50.0** |
| Ours (Framework) | 36.8 | 0.58 | 0.59 | 0.54 | *92.5* | 37.5 | 0.51 | 0.53 | 0.60 | *70.0* |

tion shifts. As shown in Table 4, most baselines collapse in OOD settings, with CWR values near 100% due to overfitting (domain-specific models) or distribution-tied rewards (RLVR). In contrast, our framework transfers uncertainty signals across domains, substantially reducing confident errors (e.g., Technique: **50%** vs. 100% for all baselines) while maintaining competitive accuracy.

> *Takeaway*
>
> Main results show that self-awareness enhances reliability: it identifies knowledge boundaries to avoid naïve RAG errors, gauges confidence to reduce harmful high-confidence mistakes, and sustains accuracy comparable to the base model.

### 4.2.2 FRAMEWORK ANALYSIS

This section examines the proposed framework from three perspectives: ablation studies, confidently wrong rate analysis, and the reward dynamics observed during training.

**Ablation Study** We evaluate four variants of our framework: (1) the full model with both boundary- and confidence-awareness rewards plus adaptive strategies (AS); (2) the self-aware LLM without AS; (3) without the boundary-awareness reward; and (4) without the confidence-awareness reward. Note that when removing one of the rewards, we do not add AS, since the adaptive strategies rely on awareness signals derived from both rewards and cannot be meaningfully tested in isolation. As shown in Figure 4, the full framework (blue) and the w/o AS variant (green) achieve the strongest overall performance on both in-distribution and out-of-distribution evaluations. The

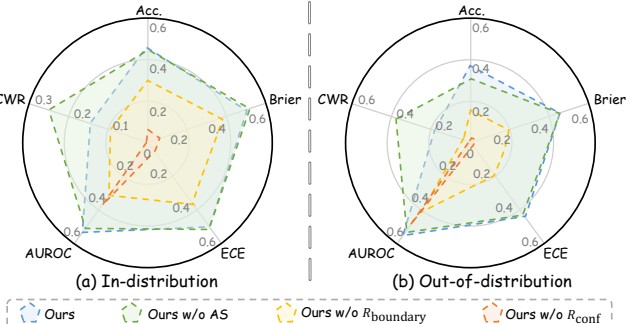

Figure 4: Radar plots of ablation study results averaged over in-distribution (left) and out-of-distribution (right) benchmarks. 'AS' refers to 'Adaptive Strategies' and 'w/o' means 'without'. For consistency across metrics, lower-is-better metrics (Brier score, ECE, and CWR) are shown as $1 - \text{metric}$.

w/o AS model achieves lower confidently wrong rate (CWR), while the full framework benefits from higher task accuracy, reflecting the complementary roles of awareness learning and adaptive strategies. By contrast, removing the confidence reward (orange) leads to severe degradation. We attribute this to the fact that a Brier-style confidence signal alone is insufficient to capture confidence reliably. Prior work (Damani et al., 2025) combines such calibration objectives with correctness-based rewards, whereas our design uses purely awareness-driven penalties. Without correctness feedback, the model struggles to form useful confidence signals. Finally, removing the boundary-

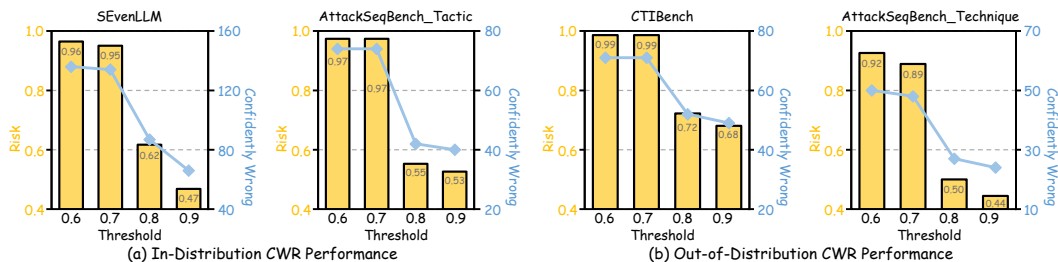

Figure 5: The confident wrong rate (CWR@$\tau$) at different thresholds $\tau$.

awareness reward (red) yields lower accuracy, as the model lacks explicit guidance to distinguish in- from out-of-boundary queries. Since this reward is the only signal explicitly tied to knowledge scope, its absence limits the model's ability to balance accuracy and faithfulness.

**Analysis of Faithfulness Under Uncertainty**  Figure 5 reports confidently wrong rate (CWR) across different confidence thresholds, complementing the results in Tables 3 and 4. At loose thresholds (0.6–0.7), our model still produces a non-negligible fraction of confidently wrong outputs. However, once the threshold is tightened (0.8–0.9), CWR drops sharply in both in-distribution and out-of-distribution settings, unlike baselines, which remain near 100%. These results reinforce our earlier findings that the framework substantially reduces highly confident errors, a property especially critical in high-stakes domains.

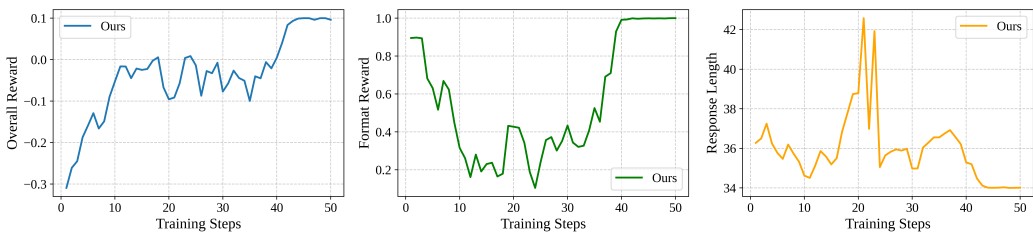

Figure 6: The three panels show the evolution of (left) overall reward, (middle) format reward, and (right) response length over training steps.

**Training Dynamics**  Figure 6 shows the training dynamics of our self-aware LLM. The left panel reports the **overall reward**, which steadily increases as training progresses, demonstrating that the model is successfully optimizing toward the self-awareness objectives. The middle panel tracks the **format reward**, which fluctuates early on but eventually converges near 1.0, indicating that the model learns to reliably produce outputs in the desired format and can subsequently focus on optimizing awareness signals. The right panel presents the **response length**, which stabilizes after initial fluctuations. The model ultimately generates shorter and more concise responses—a desirable property for efficiency, though in high-stakes settings this raises an interesting future direction: complementing concise outputs with richer reasoning or post-hoc self-explanations to strengthen trust and interpretability.

## 5  CONCLUSION

In this work, we present a new paradigm that elevates faithfulness to a first-class objective and realize it through a framework that decouples self-awareness learning from adaptive strategies, yielding clearer training objectives and more flexible inference-time control. To equip LLMs with self-awareness, we design reinforcement learning objectives that cultivate two complementary capabilities, *boundary-awareness* and *confidence-awareness*, and pair them with adaptive strategies that operationalize these signals into behavior. Extensive experiments on cybersecurity benchmarks demonstrate that self-awareness substantially reduces confidently wrong errors while maintaining competitive task performance. These results highlight that treating faithfulness as a first-class objective offers a practical path toward safer, more reliable, and ultimately more trustworthy deployment of LLMs in high-stakes domains.

## ETHICS STATEMENT

Our work aims to improve the reliability of large language models in high-stakes domains such as cybersecurity, finance, and healthcare by reducing confidently wrong outputs. While this contributes to safer deployment, potential ethical risks remain. More faithful models may still be misapplied if used without appropriate oversight, and benchmarks in specialized domains may encode biases or incompleteness. In addition, training and reinforcement learning incur computational and environmental costs. We encourage responsible use of our methods, transparent reporting of limitations, and continued efforts toward fairness, safety, and sustainability in real-world deployment.

## REPRODUCIBILITY STATEMENT

We provide an anonymous repository at https://anonymous.4open.science/r/SelfAwareLLM, which includes the full implementation used in our experiments. The paper and appendix further describe the model architecture, training objectives, and evaluation protocols in detail. These resources enable independent reproduction of our results and verification of the reported findings.

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

## A  THE USE OF LARGE LANGUAGE MODELS

We declare that generative AI tools (e.g., ChatGPT) were employed in the preparation of this manuscript. Their use was limited to grammar checking, language polishing, and enhancing the clarity and fluency of the text. In addition, they were applied in a minor capacity for debugging and syntactic correction of code snippets included in the work.

## B  EXPERIMENTAL DETAILS

### B.1  CALCULATION OF EVALUATION METRICS

The calculation of the calibration evaluation metrics are defined as follows:

$$\text{Brier} = \frac{1}{N} \sum_{i=1}^{N} (p_i - y_i)^2,  \tag{6}$$

$$\text{ECE} = \sum_{m=1}^{M} \frac{|B_m|}{N} \Big| \text{acc}(B_m) - \text{conf}(B_m) \Big|,  \tag{7}$$

$$\text{AUROC} = \frac{1}{|P||N|} \sum_{i \in P} \sum_{j \in N} \mathbf{1}\{ p_i > p_j \},  \tag{8}$$

where $p_i$ denotes the predicted probability of the positive class for instance $i$, $y_i \in \{0, 1\}$ is the ground-truth label, $B_m$ is the set of samples in the $m$-th confidence bin, $\text{acc}(B_m)$ and $\text{conf}(B_m)$ are the empirical accuracy and mean confidence within $B_m$, and $P$ and $N$ are the sets of positive and negative instances, respectively.

### B.2  DATASET DETAILS

The detailed statistics of in-boundary and out-boundary samples from cybersecurity benchmarks are shown in Table 5.

Table 5: Cybersecurity benchmark statistics

| Dataset | Total | In-Boundary | Out-of-Boundary |
|---|---|---|---|
| CSEBench | 10385 | 9558 | 827 |
| CTIBench | 2500 | 2149 | 351 |
| SEvenLLM | 5995 | 5202 | 793 |
| AttackSeq_Tactic | 1697 | 1211 | 486 |
| CyberMetric | 10200 | 9496 | 704 |
| AttackSeq_Technique | 1917 | 1655 | 262 |
| Overall | 32694 | 29271 | 3423 |

The detailed statistics of in-distribution (training data) and out-of-distribution samples are reported in Table 6. To ensure the model can successfully distinguish between in- and out-of-boundary queries, we adopt a 1:1 split for training, validation, and testing. This balanced design prevents the reinforcement learning agent from exploiting data imbalance (e.g., always predicting one type to maximize reward), which could otherwise yield deceptively good performance without reflecting genuine boundary-awareness.

### B.3  ADDITIONAL RESULTS

We evaluate four variants of our framework: (1) the full model with both boundary- and confidence-awareness rewards plus adaptive strategies (AS); (2) the self-aware LLM without AS; (3) without the boundary-awareness reward; and (4) without the confidence-awareness reward. When removing

Table 6: Cybersecurity benchmark statistics

| | Dataset | Train (in/out) | Valid (in/out) | Test (in/out) |
|---|---|---|---|---|
| In-Distribution | CSEBench | 1240 (620/620) | 164 (82/82) | 250 (125/125) |
| | SEvenLLM | 1188 (594/594) | 158 (79/79) | 240 (120/120) |
| | AttackSeq_Tactic | 728 (364/364) | 96 (48/48) | 148 (74/74) |
| | CyberMetric | 1056 (528/528) | 140 (70/70) | 212 (106/106) |
| Out-of-Distribution | AttackSeq_Technique | - | - | 80 (40/40) |
| | CTIBench | - | - | 106 (53/53) |

one of the rewards, we do not include AS, since adaptive strategies depend on awareness signals derived from both rewards and cannot be meaningfully tested in isolation.

As shown in Table 7 and summarized in Figure 4, the full framework (blue) and the w/o AS variant (green) deliver the strongest performance across both in-distribution and out-of-distribution settings. The w/o AS variant achieves lower confidently wrong rates (CWR), while the full framework provides higher accuracy, highlighting the complementary roles of awareness learning and adaptive strategies.

By contrast, removing the confidence-awareness reward (orange) results in severe degradation. This is because a Brier-style confidence penalty alone cannot capture reliable awareness signals. Prior work (Damani et al., 2025) mitigates this by mixing calibration with correctness-based rewards, whereas our design employs purely awareness-driven objectives. Without correctness feedback, the model struggles to form meaningful confidence estimates.

Finally, removing the boundary-awareness reward (red) leads to consistently lower accuracy, as the model lacks explicit guidance to distinguish in- from out-of-boundary queries. Since this reward is the only one tied directly to knowledge scope, its absence prevents the model from balancing accuracy with faithfulness.

Table 7: Results of ablation study.

| Benchmarks | In-Distribution | | | | | | | | | |
|---|---|---|---|---|---|---|---|---|---|---|
| | CSEBench | | | | | CyberMetric | | | | |
| Metrics | Acc. ↑ | Brier ↓ | ECE ↓ | AUROC ↑ | CWR@0.8 ↓ | Acc. ↑ | Brier ↓ | ECE ↓ | AUROC ↑ | CWR@0.8 ↓ |
| Ours | 0.444 | 0.5541 | 0.5552 | 0.4981 | 1 | 0.4811 | 0.5039 | 0.5023 | 0.5021 | 0.9545 |
| Ours w/o AS | 0.44 | 0.5469 | 0.5494 | 0.5151 | 0.95 | 0.4764 | 0.4815 | 0.4798 | 0.5206 | 0.8559 |
| Ours w/o $R_{boundary}$ | 0.024 | 0.98 | 0.98 | 0.3354 | 0.9959 | 0.1368 | 0.8726 | 0.8726 | 0.451 | 0.9945 |
| Ours w/o $R_{conf}$ | 0.332 | 0.5963 | 0.6026 | 0.3822 | 0.9341 | 0.3443 | 0.5631 | 0.5682 | 0.4235 | 0.9137 |

| Benchmarks | In-Distribution | | | | | | | | | |
|---|---|---|---|---|---|---|---|---|---|---|
| | SEvenLLM | | | | | AttackSeqBench_Tactic | | | | |
| Metrics | Acc. ↑ | Brier ↓ | ECE ↓ | AUROC ↑ | CWR@0.8 ↓ | Acc. ↑ | Brier ↓ | ECE ↓ | AUROC ↑ | CWR@0.8 ↓ |
| Ours | 0.3958 | 0.4938 | 0.5106 | 0.5912 | 0.7724 | 0.5473 | 0.3798 | 0.3828 | 0.6181 | 0.6866 |
| Ours w/o AS | 0.4125 | 0.4671 | 0.4682 | 0.539 | 0.617 | 0.4865 | 0.4121 | 0.4244 | 0.6059 | 0.5526 |
| Ours w/o $R_{boundary}$ | 0.0375 | 0.9798 | 0.9808 | 0.2232 | 0.9957 | 0 | 1 | 1 | 0.5 | 1 |
| Ours w/o $R_{conf}$ | 0.3292 | 0.5542 | 0.5526 | 0.3503 | 0.8137 | 0.2162 | 0.7374 | 0.734 | 0.188 | 0.9569 |

| Benchmarks | Out-of-Distribution | | | | | | | | | |
|---|---|---|---|---|---|---|---|---|---|---|
| | CTIBench | | | | | AttackSeqBench_Technique | | | | |
| Metrics | Acc. ↑ | Brier ↓ | ECE ↓ | AUROC ↑ | CWR@0.8 ↓ | Acc. ↑ | Brier ↓ | ECE ↓ | AUROC ↑ | CWR@0.8 ↓ |
| Ours | 0.3679 | 0.5807 | 0.5904 | 0.5434 | 0.9254 | 0.375 | 0.5058 | 0.5284 | 0.5983 | 0.7 |
| Ours w/o AS | 0.3208 | 0.5865 | 0.6113 | 0.578 | 0.7222 | 0.325 | 0.5018 | 0.534 | 0.5962 | 0.5 |
| Ours w/o $R_{boundary}$ | 0.0189 | 0.9811 | 0.9811 | 0.5 | 1 | 0 | 1 | 1 | 0.5 | 1 |
| Ours w/o $R_{conf}$ | 0.2925 | 0.6085 | 0.6133 | 0.3262 | 0.9067 | 0 | 1 | 1 | 0.5 | 1 |

## B.4 PROMPTS

The prompt for estimating the knowledge boundary from Section 3.1 is given in Prompt B.1. The system prompt for training and evaluating out method is given in Prompt B.2

---

**Prompt B.1: Estimation of Model's Knowledge Boundary of CTI Prompt**

You are a cybersecurity expert specializing in Cyber Threat Intelligence (CTI) and adversary behaviors, with strong knowledge of frameworks such as MITRE ATT&CK, the CWE database, and CAPEC patterns.

Your task is to determine which answer choice is the most accurate and plausible response to the given question.

Question:
{question}
Options:
{options_text}

Please conclude with your final answer in the following format: "Final Answer: `<insert answer choice here>`"

---

**Prompt B.2: Estimation of Model's Knowledge Boundary of CTI Prompt**

You are a cybersecurity expert specializing in Cyber Threat Intelligence (CTI) and adversary behaviors, with deep knowledge of frameworks such as MITRE ATT&CK, the CWE database, and CAPEC patterns.

Your task is to analyze the given question and its answer options, then select the most accurate and plausible choice.

Instructions
1. Carefully evaluate each option using your expertise.
2. Select the single best answer. Only use the provided option labels (e.g., A, B, C, D).
3. Do not include reasoning or explanations in the output.
4. Format your final response exactly as follows:
`<answer> Final Answer:  [answer here] </answer>`
`<confidence> number between 0 and 100 </confidence>`
`<rag> True or False </rag>`

Confidence Rating Rules
- 90–100: Absolute certainty; the answer is definitely correct.
- 70–89: High confidence; very likely correct, but not absolute.
- 50–69: Moderate confidence; likely correct, but some uncertainty due to missing details.
- 30–49: Low confidence; educated guess with limited supporting knowledge.
- 0–29: Very low confidence; mostly a guess.

RAG Determination Rules
- True → Additional knowledge required or question ambiguous
- False → Question is clear and can be answered with internal knowledge

Consistency Rule
- If `<rag>` is True, confidence should usually be lower (typically below 70), since uncertainty or missing knowledge is implied.
- If `<rag>` is False, confidence can range freely, including high values, depending on certainty.

Question:
{question}

Options:
{options_text}

---

