# OpenReview forum: "Equipping LLMs with Self-Awareness for High-Stakes Tasks"
_ICLR.cc/2026/Conference — Submitted to ICLR 2026_

### Official Review · Reviewer_LaYM · 2025-10-26

**Soundness:** 3
**Presentation:** 2
**Contribution:** 2
**Rating:** 4
**Confidence:** 4

**Summary:**

The paper highlights the importance of enhancing LLM self-awareness in high-stakes domains.
It proposes a novel reinforcement learning framework that incorporates two awareness-based rewards, boundary-awareness and confidence-awareness, together with adaptive inference-time strategies.
Experimental results show that it significantly reduces the proportion of confidently wrong outputs.

**Strengths:**

- Equipping LLMs with _self-awareness_ for **high-stakes applications** is practically meaningful and timely.
- Using reinforcement learning purely with awareness rewards, without optimizing correctness, is a novel and thought-provoking exploration.
- The experiments are relatively comprehensive, covering 4 datasets and 5 metrics.

**Weaknesses:**

- The paper introduces new notions (self-awareness, boundary-awareness, confidence-awareness) but lacks definitions and fails to clarify how they relate to areas like _confidence calibration_ or uncertainty estimation.
- The conceptual distinction between “what it knows” and “how well it knows” is unclear and not empirically separated. If confidence estimation is accurate, it could already imply boundary awareness, so the necessity of decoupling these two objectives is not well-justified.
- Experimental results (Table 3) show that the proposed method underperforms strong baselines such as **Primus-Merged** and **RLVR** on most standard metrics (Accuracy, Brier, ECE, AUROC), making it difficult to claim clear advantages.
- The proposed **CWR** metric is highly sensitive to threshold choice. Improvements appear only at high thresholds (≥ 0.8); the paper does not explain why this threshold is chosen or how many samples fall into this region.
- Important calibration baselines (e.g., verbalization[1], self-consistency[2]) are missing, which weakens the empirical comparison.
- The IB/OOB classification based on “at least one correct sample” is questionable. This can be affected by sampling frequency or prompt design.

[1] Just Ask for Calibration: Strategies for Eliciting Calibrated Confidence
Scores from Language Models Fine-Tuned with Human Feedback
[2] Selfcheckgpt: Zero-resource black-box hallucination detection for generative large language models.

**Questions:**

see above

---

> ### Author Response · Authors · 2025-11-22
> **Response to Reviewer LaYM**
>
> We sincerely thank you very much for your valuable time and constructive review! We address your questions and concerns below:
>
> - Response to Weakness #1: We will add concise formal definitions in the Method section and clarify the link to calibration/uncertainty.
>     - **Boundary-awareness** refers to the model’s ability to judge whether an input is in- vs. out-of-scope of its knowledge boundary (closely related to knowledge-boundary literature).
>     - **Confidence-awareness** refers to the model’s verbalized confidence reflecting its likelihood of being correct, which connects to uncertainty estimation and confidence calibration.
>     - **Self-awareness** in our paper denotes the joint ability of boundary-awareness and confidence-awareness, operationalized for high-stakes decision support rather than as a new theoretical construct.
> - Response to Weakness #2: While, in theory, perfectly calibrated confidence could implicitly capture knowledge boundaries, achieving such calibration is difficult in large language models. Given their well-known tendency toward overconfidence [3], relying solely on confidence is insufficient. Boundary-awareness provides an explicit corrective signal by encouraging the model to recognize when correctness is unattainable, regardless of its confidence level.
>
>     [3] Why Language Models Hallucinate
>
> - Response to Weakness #3: Thank you for pointing this out. While baselines such as Primus-Merged and RLVR achieve stronger overall task performance, partly due to domain-specific supervision through explicit reward shaping and SFT, and the Primus versions injects a lot of domain-specific knowledge. Our framework is specifically designed to reduce **confidently-wrong errors**, which represent the most critical failure mode in high-stakes decision-making settings. Accordingly, we evaluate and emphasize CWR, where our method achieves a substantial improvement relative to these baselines. In other words, our contribution is not higher average accuracy or generic calibration, but a targeted reduction of **high-confidence mistakes** (tail-risk).
> - Response to Weakness #4: Conceptually, CWR is designed to measure instances where the model is wrong yet highly confident. A threshold around 0.8 serves as a practical boundary between “acceptable confidence” and “overconfidence.” Empirically, as shown in Figure 5, **our model exhibits consistent CWR trends across different threshold settings**, with the most notable reduction observed at 0.9. We will clarify this rationale and its interpretive basis in the revised manuscript.
> - Response to Weakness #5: We appreciate the reviewer’s comment and fully agree that [1] and [2] are excellent calibration techniques focusing on improving confidence estimation accuracy. However, our work does not aim to propose another calibration method. Instead, **we develop an end-to-end decision framework designed for high-stakes scenarios**, where calibration serves as one component rather than the final objective. Hence, our experimental comparisons focus on task-oriented, high-stakes decision frameworks (e.g., RLVR, Primus) that share similar goals.
>
>     [1] Just Ask for Calibration: Strategies for Eliciting Calibrated Confidence Scores from Language Models Fine-Tuned with Human Feedback
>     [2] Selfcheckgpt: Zero-resource black-box hallucination detection for generative large language models.
>
> - Response to Weakness #6: Thank you for raising this. Identifying a model’s knowledge boundary is inherently challenging, as there is no agreed-upon formal definition in prior work. Our heuristic, classifying an input as in-boundary (IB) if at least one of multiple sampled answers is correct, offers a practical and empirically stable approximation.
>     - On sensitivity to sampling frequency: In preliminary experiments, we varied the sampling number $N \in [8，16，32，64]$. As $N$ increases, the proportion of samples labeled as “OOB” (all incorrect among $N$ trials) decreases slightly and stabilizes as $N = 16$, indicating that $N=16$ provides a reasonable trade-off between stability and cost. The distribution of correct counts is bimodal, mainly concentrated at $N/N$ (clearly in-boundary) and $0/N$ (out-of-boundary). Samples with only a few correct trials (e.g., 1–2) are rare and have limited effect on boundary quality.
>     - On prompt dependence: To control for prompt-related variation, **we use consistent prompt formats between training and testing** (except for the confidence component) across all methods, including baselines. Furthermore, we evaluated multiple paraphrased prompts for the same queries and observed minimal changes in IB/OOB classification. These results suggest that our boundary identification remains consistent under moderate prompt variations.

---

> ### Author Response · Authors · 2025-11-22
> **Summary**
>
> Thank you again for the careful reading and constructive feedback, and for recognizing the strengths of our work. Our work aims to empowering LLMs with self-awareness for high-stakes tasks: we identify **self-awareness (knowledge-boundary and confidence-awareness)** as the missing capability behind overconfidence, and propose— to our knowledge, for the first time—an **RL framework that trains this self-awareness by decoupling awareness learning from task correctness**, then pairing it with adaptive inference strategies (e.g., RAG and low-confidence regeneration). This yields large, consistent reductions in confidently wrong outputs.
>
> In the above responses, we have addressed your concerns point-by-point. If you have any further questions, please let us know—we would be happy to clarify. **If there are no remaining concerns, we kindly ask you to reconsider your evaluation in light of these clarifications.**

---

> ### Comment · Reviewer_LaYM · 2025-11-25
>
> Thank you for the response.
> However, the rebuttal does not fully resolve my concerns regarding the soundness and contribution of the proposed method.
>
> The authors acknowledge that the approach underperforms strong baselines on standard and widely accepted metrics such as Accuracy, ECE, and AUROC. Given this, relying primarily on the proposed CWR metric is insufficient to justify the degradation in overall performance and calibration quality.
> In addition, the distinction between Boundary-awareness and Confidence-awareness is still not clearly articulated, and the motivation for treating them as separate objectives has not been substantiated.
>
> The rationale for excluding established calibration baselines remains unconvincing. Since "confidence-awareness" is presented as a central component of the framework, comparisons with state-of-the-art calibration or hallucination-detection methods would be necessary to demonstrate the necessity and advantage of the proposed RL-based solution over simpler inference-time techniques.
>
> Thus, I decide to keep my score.

---

### Official Review · Reviewer_NfsF · 2025-10-28

**Soundness:** 4
**Presentation:** 3
**Contribution:** 3
**Rating:** 8
**Confidence:** 4

**Summary:**

The paper introduces an approach to explicitly train models to recognize the limits of what they know and to assess how well they know it (i.e., assigning confidence). The main innovation is considering this as a separate task from providing the model with more knowledge. Experiments are conducted on 6 benchmarks from the cybersecurity domain and compared with several baselines.

**Strengths:**

- Originality: the approach proposed by the authors is simple and elegant, but most likely original
- quality: the way in which the method is framed and tested is of high quality
- clarity: the explanation of the method and presentation of the results is mostly clear
- signitficance: the paper’s main signiicance is in arguing that “faithfulness must be treated as a first-class objective rather than a by-product of knowledge expansion”, beyond the specifics of the implemented method.

**Weaknesses:**

The paper has a few minor weaknesses:

- overselling the significance of the results: it is true that the method proposed by the authors has good performance, comparable or better than the other baselines. However, even with this method, the scores for AUC, ECE, and CWR@0.8 are, in absolute terms, bad (ie, the models have still poor confidence quantification abilities!). This does *not* reduce the significance of the authors’ work, as the limited strength of a method on the existing set of models should not work against how worth of publication the method is (particularly when the method is comparable to other baselines). But I invite the authors to not oversell the results explicitly or implicitly, such as by saying “enhances reliability” without stressing that this is still poor as this may have dangerous consequences, for instance inducing practitioners to adopt models that are not suitable in high-stakes scenarios.
- a few sentences are unclear:
    - the opening in the abstract ‘Overconfidence in large language model responses’: not clear if overconfidence of the models or users
    - line 68 is unclear: “[figure 1b] shows that…”
    - line 69: ‘ordinary hallucinations’: it would be good to define this.
- the authors introduce a new term ‘self-awareness’, but meta-cognition is used in the cognitive sciences for the same aim. Is there need to introduce such additional term?
- the related works section can touch a few additional points:
    - papers discussing fundamental limits on LLM’s confidence and correctness, as for instance https://arxiv.org/abs/2408.02357, https://arxiv.org/abs/2509.04664
    - works to measure the ability to determine high-confidence and high-correctness regions for high-stakes AI usage, such as https://aclanthology.org/2025.findings-acl.790/
    - empirical measures of overconfidence: https://www.nature.com/articles/s41586-024-07930-y
    - the ‘assessor’ class of methods using surrogate models to estimate uncertainty: https://scholar.google.com/citations?view_op=view_citation&hl=en&user=VV76Eo8AAAAJ&citation_for_view=VV76Eo8AAAAJ:Y0pCki6q_DkC
- while the experiments include OOD datasets, it would be interesting to see how the methods affects completely different domains. Moreover, I would be interesting in seeing how this training method, that is likely applied using benchmarks with well-defined ground-truth questions, affects the model’s behaviour on more open-ended tasks where a single ground truth may not exist.

**Questions:**

- is it fair to say that a very high ability to “know how well it knows” implies “know what it knows?”. Or that, put otherwise, the former cannot exist without the second?
- Table 1 mentions that, if the model determines a question is out of its knowledge boundary, it shoudl use RAG. But what if there is nothing that is not relevant to the quesiton at hand in the knowledge database?
- Can one applied the suggested approach to a surrogate (smaller) model that accompanies a larger one which is tasked with producing the actual answer?
- Do the authors have any intuition on what the effect would be on completeyl different domains, or on open-ended tasks with no single ground truth? Ie, does the model learn its knowledge boundary and confidence for the specific considered domain, or does the training method unlocks an “introspection” capability (as in https://arxiv.org/abs/2410.13787)?

---

> ### Author Response · Authors · 2025-11-22
> **Response to Weaknesses (#1 - #5)**
>
> Thank you very much for the positive and encouraging review, and for recognizing the contribution of our work. We greatly appreciate your thoughtful feedback and constructive suggestions. Below we address the concerns and questions you raised.
>
> - Response to Weakness #1: Thank you for raising this point. We agree that the absolute AUC/ECE/CWR values show confidence quantification is still far from solved for current LLMs. Our claim is therefore relative, not absolute: the framework mainly reduces confidently-wrong errors compared to strong baselines, which is the risk that matters most in high-stakes settings. We will tune down wording such as “enhances reliability,” explicitly state that calibration remains imperfect, and add a brief limitation/risk note to prevent over-interpretation of deployment readiness.
> - Response to Weakness #2: Thank you for these detailed points. We will revise the manuscript accordingly.
> - Response to Weakness #3: We agree that “meta-cognition” is the established term in cognitive science. We introduce “self-awareness” only as an **LLM-safety operationalization in high-stakes settings**: a decision-oriented capability where the model aligns its verbalized confidence and in/out-of-boundary judgment with correctness under explicit scope constraints. The term is used to emphasize this concrete, deployable objective for LLM decision pipelines, rather than to propose a new cognitive construct. In the revision, we will explicitly map “self-awareness” to meta-cognition and state that our usage is a scoped, high-stakes instantiation of that concept.
> - Response to Weakness #4: We will expand Related Work to cover the suggested directions:
>     - **Limits on confidence/correctness and hallucinations:** we will cite and discuss CRP-style limits on consistency vs. fallibility and recent analyses of why hallucinations persist in modern training pipelines.
>     - **High-confidence/high-correctness regions for high-stakes usage:** we will add the PredictaBoard benchmark and relate it to our goal of reducing high-confidence errors.
>     - **Empirical overconfidence evidence:** we will incorporate recent empirical findings showing systematic LLM overconfidence and its risks.
>     - **Assessor/surrogate-based uncertainty methods:** we will add this line of work and contrast it with our text-only, end-to-end awareness learning.
> - Response to Weakness #5: We agree that this is an important extension.
>     - To evaluate cross-domain generalization, we **additionally applied our framework to a medical domain**, another high-stakes setting. We are currently running these experiments and will include the results by the end of the rebuttal period.
>     - For open-ended tasks without a single ground truth, boundary definitions are inherently less unambiguous; we will add a limitation note stating that adapting boundary-awareness to such settings is non-trivial and left for future work.

---

> ### Author Response · Authors · 2025-11-22
> **Response to Questions (#1 - #4)**
>
> - Response to Question #1: This is a really insightful question. In principle, a perfectly calibrated model might implicitly reflect its knowledge boundary through confidence. In practice, however, LLMs are systematically overconfident and poorly calibrated, so “knowing how well it knows” is not reliable enough to recover “what it knows.” Our boundary-awareness is introduced exactly as an explicit, orthogonal signal: it teaches the model to recognize cases where correctness is unattainable regardless of its current confidence.
> - Response to Question #2: This is an excellent and very practical edge case, thank you for raising it. Our framework treats RAG as a best-effort recovery step when the model believes the query is out-of-boundary. If retrieval returns no relevant evidence (or only weakly related passages), the framework does not force a confident answer. That is, RAG only helps when such support actually exists. We will clarify this failure-mode handling in the revision.
> - Response to Question #3: This is a great and practical question. In principle, a surrogate can be used—prior work has explored small probes/classifiers or auxiliary LLMs for uncertainty/routing [1]. However, in our setting a separately trained surrogate is not ideal because knowledge boundary and confidence are model-specific; two models can disagree, yielding inconsistent decisions. This mismatch can further grow since our RL training reshapes the answer model over time. A more viable variant would be to train/distill the surrogate jointly end-to-end with the answer model, which we see as a promising extension left for future work.
>
>     [1]. Beyond Binary Rewards: Training LMs to Reason About Their Uncertainty
>
> - Response to Question #4: This is a very helpful question, and we appreciate the reviewer for connecting our setup to the broader “introspection” perspective. Our intuition is that the two parts of our framework transfer differently. Confidence-awareness targets how the model self-reports uncertainty, and similar to standard calibration, we expect this skill to be relatively domain-agnostic. Boundary-awareness, however, is tied to how a domain defines “in-boundary vs. out-of-boundary” knowledge; thus its benefits are strongest when the boundary is operationally meaningful and reasonably well-specified (as in high-stakes, knowledge-intensive domains). For open-ended tasks with no single ground truth, the notion of a crisp knowledge boundary becomes inherently blurred, so our current boundary-aware objective is less directly applicable. In those settings, traditional calibration approaches that do not rely on explicit boundary structure may be a better fit. Accordingly, we do not claim universal open-domain introspection here; rather, we view our method as enabling targeted introspection for high-stakes domains, and extending it to open-ended/open-domain settings is an interesting direction for future work.

---

> ### Author Response · Authors · 2025-11-22
> **Summary**
>
> Thank you again for the positive and constructive review. Our work targets high-stakes settings by equipping LLMs with self-awareness, which we formalize as knowledge-boundary awareness and confidence awareness. We hope the responses above address your concerns, and we appreciate any further feedback.

---

### Official Review · Reviewer_zAPH · 2025-10-31

**Soundness:** 2
**Presentation:** 3
**Contribution:** 2
**Rating:** 0
**Confidence:** 2

**Summary:**

This paper introduces a framework for “equipping large language models with self-awareness,” proposing reinforcement learning objectives that explicitly train two complementary capabilities: boundary-awareness and confidence-awareness. The central idea is to optimize for awareness signals—the model’s ability to recognize when it knows or does not know—rather than directly for correctness. The resulting “self-aware” LLM can avoid producing confidently wrong answers even when uncertain. Crucially, the paper separates awareness learning (trained independently of task incentives) from downstream inference-time strategies, such as retrieval-augmented generation and low-confidence regeneration. Experiments on cybersecurity benchmarks—a domain chosen for its high stakes and data scarcity—show that the framework significantly reduces confidently wrong outputs compared to strong baselines, while maintaining competitive accuracy. When combined with the adaptive inference strategies, the method further improves robustness and task performance, suggesting that explicitly modeling self-awareness offers a principled route to safer model deployment in critical settings

**Strengths:**

The paper identifies an important and underexplored goal—reducing confidently wrong model behaviors—by directly training self-awareness instead of relying on external calibration or confidence post-processing. The conceptual framing is clear and well-motivated: awareness learning as an independent signal decoupled from correctness offers a theoretically elegant way to improve reliability without distorting model incentives. The implementation is also thoughtfully designed: reinforcement learning objectives are introduced for boundary-awareness and confidence-awareness, and these are combined with adaptive inference-time mechanisms that modulate generation based on the learned signals. The experiments are extensive within their chosen domain, covering both in-distribution and out-of-distribution cybersecurity datasets.

**Weaknesses:**

While the paper presents self-awareness as broadly applicable, the practicality of the approach is limited by its dependence on internal confidence scores and gradient-based access, which are often unavailable in real deployment settings. The framework assumes the ability to read and train confidence-bearing logits, compute awareness rewards, and integrate custom reinforcement learning objectives. However, many widely used LLMs—especially proprietary API-based models—do not expose raw probabilities, hidden activations, or confidence estimates in a stable or well-defined manner. Even when approximate probabilities are available, they are frequently temperature- or decoding-dependent and thus not reliable indicators of epistemic uncertainty. As a result, the proposed approach cannot be applied to many realistic scenarios where organizations rely on closed-weight models that allow only text-in/text-out interaction.

Another concern is that the paper treats confidence as a coherent and interpretable signal, yet the relationship between logit magnitude and genuine model uncertainty is not theoretically grounded. Modern LLMs often produce overconfident distributions due to training dynamics unrelated to knowledge boundaries, and reinforcement training on confidence signals may amplify this effect rather than mitigate it. The paper claims that its awareness objectives produce a “pure” measure of model uncertainty, but provides no mechanistic or representational analysis to support this, nor does it examine whether the trained score aligns with human judgments of uncertainty. Without probing how or where this awareness is encoded in the model’s internal representations, it remains unclear whether the system is genuinely learning to recognize when it does not know something, or simply learning a new pattern of confidence suppression conditioned on dataset artifacts.

A further limitation is that the evaluation design does not reflect natural failure modes. The cybersecurity datasets are constructed with explicit in-boundary and out-of-boundary distinctions, creating a relatively sharp and learnable separation in input space. Many real-world uncertainty cases—e.g., ambiguous facts, partially overlapping knowledge domains, multi-step reasoning failures—do not exhibit such clean boundaries, and the method’s performance under these more subtle conditions is not demonstrated. The balanced train/test splits further simplify the decision setting and may encourage the model to rely on distribution memorization rather than genuine introspection.

The inference-time adaptive strategies—retrieval and low-confidence regeneration—likely contribute significantly to the observed improvements, yet the paper does not disentangle their effects from those of the self-awareness rewards. Retrieval-augmented generation, in particular, already encourages models to hedge or reconsider initial claims when uncertain. If most of the benefit comes from retrieval or regeneration, then the core contribution—self-awareness learning—may be less central than claimed. The existing ablations gesture toward this question but do not isolate it convincingly.

**Questions:**

How robust are the learned awareness signals when applied to tasks beyond cybersecurity, particularly in open-domain reasoning or dialogue generation, where uncertainty manifests differently?

Can the authors provide evidence that the awareness signals correspond to interpretable internal confidence or epistemic uncertainty, rather than being indirect by-products of reward optimization?

---

> ### Author Response · Authors · 2025-11-22
> **Response to Reviewer zAPH**
>
> - Response to Weakness #1 & 2: We respectfully note that several of the reviewer’s concerns stem from an incorrect assumption that our method defines or trains confidence based on logits or internal uncertainty signals. This is not the case.
>     - Our framework does not access logits, hidden activations, internal probability distributions, or any gradient-based confidence scores. All confidence signals used for training are derived exclusively from verbalized confidence, explicit natural-language self-reports produced by the model itself. The awareness reward is computed solely over these text outputs.
>     - Accordingly, critiques involving logit magnitudes, temperature-dependent softmax distributions, or reinforcement learning potentially amplifying logit-based overconfidence are not applicable to our setting. Our approach does not rely on logits as a source of epistemic information, and we make no theoretical assumptions linking logit magnitude to genuine uncertainty. Confidence in our framework is a behaviorally defined, text-level signal, evaluated with correctness-conditional calibration metrics rather than internal probability values.
>
>     - Regarding applicability to closed-weight models, we agree that training any tuning-based method, including ours, requires access to model parameters. This limitation is shared by all RL- and SFT-based approaches and is not specific to our framework.
>     - Finally, the reviewer raises concerns about whether the learned confidence signal constitutes a “coherent” or “pure” uncertainty measure, and whether it corresponds to internal epistemic representations. We agree that our method does not establish mechanistic interpretability of awareness signals or recover an internal uncertainty variable. Nor do we claim that the model learns a representational notion of epistemic uncertainty. Our contribution is explicitly behavioral and tailored to the intended high-stakes deployment setting, where decision pipelines often rely on explicit, model-provided confidence statements rather than internal epistemic representations.
> - Response to Weakness #3: The evaluation is intentionally aligned with high-stakes domains where boundaries are operationally defined (e.g., known vs. unknown techniques, covered vs. uncovered policies). The goal is not open-domain generalization, but safer decision-making under domain-specific scope limits. The controlled boundary setup is thus a realistic instantiation of the targeted use case.
> - Response to Weakness #4: As shown in Table 3, Ours (LLM), trained with awareness learning only, already achieves a significant reduction in CWR, indicating that the awareness objectives alone effectively reduce overconfident errors. In contrast, Ours (Framework), which adds adaptive inference, shows slightly higher accuracy but higher CWR, suggesting that adaptive strategies mainly enhance task completion rather than awareness itself. This complementary pattern confirms that awareness learning and adaptive inference play distinct, non-overlapping roles.
> - Response to Question #1: As stated in the title, abstract, and introduction, the goal of this work is to improve reliability in high-stakes domains, where uncertainty has concrete operational meaning and the cost of confident errors is substantial. Our evaluations therefore focus on such domains by design. Extending the framework to open-domain reasoning is a valuable direction but lies outside the intended scope of this work.
> - Response to Question #2: The question seems to assume that our setting involves internal epistemic-uncertainty modeling. However, our confidence signals are entirely verbal, and the reward explicitly targets accurate self-reporting behavior, not internal uncertainty estimation. We therefore do not, and did not intend to, claim correspondence with internal uncertainty variables.

---

### Official Review · Reviewer_tSAQ · 2025-11-01

**Soundness:** 2
**Presentation:** 3
**Contribution:** 2
**Rating:** 4
**Confidence:** 3

**Summary:**

The paper proposes a self-awareness RL-tuning framework for LLMs in high-stakes domains, decomposed into boundary-awareness and confidence-awareness. It operationalizes both objectives via a binary reward for in/out-of-boundary classification and a calibration reward based on Brier score over verbalized confidence, and optimizes using GRPO with a curriculum. On six cybersecurity benchmarks, the approach substantially lowers the Confident Wrong Rate at a high confidence threshold, but gains over baselines on other calibration metrics or task accuracy are not consistent nor significant.

**Strengths:**

* Evaluating on real-world high-stake domains (in this paper, cybersecurity) is valuable (which is not often seen from previous works).
* Optimizing jointly for boundary-awareness and confidence-awareness is a clean approach, and is a promising direction for increasing trustworthiness in high-stake and critical applications, which is a significant topic.
* The ablation studies on different reward terms and inference-time strategies are comprehensive, and presented with good visualization and analysis.

**Weaknesses:**

* The authors should make it clearer their unique contributions compared with prior works. In particular, RLCR [1] which this paper cites and compares as a baseline, shares the exact same confidence-aware reward formulation, Brier score based scoring rule, and also uses GRPO for the RL algorithm. While the utility term in RLCR directly optimizes for correctness and this paper uses an indirect measure of knowledge boundary based on ensembles (additional overhead), the general goal and formulation is also similar. There is no theoretical analysis on the reward optimality or different choices of scoring functions in this paper. The authors could further elaborate on why rewarding knowledge boundary awareness instead of directly correctness could provide performance gains (based on claims from this paper), especially in OOD settings.
* While cybersecurity is certainly a representative high-stakes and knowledge-intensive domain to test on (see strengths), it would be interesting to show how this generalizes and transfers to other domains, beyond looking at distribution shifts within one domain.
* The results in both in- and out-of-distribution settings do not seem consistently strong, often not even beating the second-best, in terms of both accuracy and calibration. This is even true when augmented with adaptive strategies, which creates additional overhead at inference time, but benefits seem diminishing. Results on the self-proposed CWR metric show good gains. However, I think this metric can be flawed/gamed: 1). It can hinge on the choice of tau - what is the distribution of the overall confidence estimates, and how sensitive is the CWR when tau varies around 0.7-0.8, and 2). It may not capture the full picture: it ignores the coverage, or how often the model is highly confident - say a model is / learns to be only highly-confident for very few of all inputs (possible for difficult tasks), it can achieve a low CWR by almost never being confident at all, yet another more useful model could have the same or slightly higher CWR, but actually keep very few of its mistakes at high-confidence region.
* The in-boundary vs. out-of-boundary is determined by resembling multiple answers and see if the correct answer appears at least once. This seems to be a simplification following [2] as the paper mentioned, but the authors should elaborate on the motivations of this specific adaptation. To me, this simple approach could mis-label hard but in-boundary queries (false OOB) or lucky guesses (false IB) which induces noise, and quality might be sensitive to N and temperature. Requiring multiple samples also adds overhead (N=16 in the experiments, which seems quite significant).

---

[1]. Beyond Binary Rewards: Training LMs to Reason About Their Uncertainty

[2]. Reinforced Internal-External Knowledge Synergistic Reasoning for Efficient Adaptive Search Agent

**Questions:**

* In OOD, why does CWR improve considerably while Brier / ECE remain mixed?
* How sensitive the result is to the ensemble size N and the rollout and inference temperatures? How were the rest parameters tuned?
* The Primus series models are based on Llama-3.1, but your experiments are with Qwen-2. Did you fine-tune Qwen-2 using Primus datasets, or was the result based on Llama-3.1? What exactly is the difference between Primus-Merged and Primus-Reasoning? These details, especially a brief overview of the Primus variants, could be included when you introduce the baselines (same for SecurityLLM).

---

> ### Author Response · Authors · 2025-11-22
> **Response to Weaknesses (#1 - #4)**
>
> We sincerely thank you very much for your valuable time and constructive review! We address your questions and concerns below:
>
> - Response to Weakness #1: Thank you for the insightful comment. We share components such as the Brier-style reward and GRPO with RLCR [1], as these are well-established and effective techniques for calibration and reinforcement learning. However, despite sharing these components, **our core objective and reward formulation are fundamentally different** as below:
>     - **Why not directly optimize correctness**: As highlighted in recent work [3], training models to maximize correctness tends to incentivize “guessing” under uncertainty, leading to high-confidence errors. Avoiding such behavior in high-stakes settings is a central motivation of our work. To this end, we explicitly decouple the reward from correctness so that the model is encouraged to express uncertainty rather than to optimize for appearing correct.
>     - **Why optimize for knowledge-boundary awareness instead**: For "out-of-boundary" cases, rewarding for correctness is often illogical and counter-productive, as it punishes the model for failing to correctly answer questions that it is unlikely to know. Instead, we reward the model for the appropriate behavior in this situation: recognizing its own limitation.
>
>     Our approach empowers models with self-awareness to mitigate confidently wrong predictions and achieve more reliable behavior than correctness-driven methods.
>
>     [1] Beyond Binary Rewards: Training LMs to Reason About Their Uncertainty
>     [3] Why Language Models Hallucinate
>
> - Response to Weakness #2: We appreciate your suggestion. To evaluate cross-domain generalization, we **additionally applied our framework to a medical domain**, another high-stakes setting. We are currently running these experiments and will include the results by the end of the rebuttal period.
> - Response to Weakness #3: Thank you for this insightful observation. Figure 5 illustrate the results of $\tau \in [0.6, 0.7, 0.8, 0.9]$. In isolation, a “never-confident” model could achieve a low (better) CWR without being useful. However, **CWR is not intended to replace accuracy or calibration metrics**; it serves as a complementary safety metric. In our evaluation, we jointly consider CWR, accuracy, and calibration metrics. As shown in Table 2 & 3, our model reduces CWR while maintaining comparable accuracy (comparing with backbone, which injects no additional domain-specific knowledge) and calibration metric. Indicating that **it is not merely suppressing confidence uniformly** but learns to selectively reduce confidence on high-risk (OOD or difficult IB) inputs while retaining high confidence on clearly in-boundary cases.
> - Response to Weakness #4: Identifying a model’s knowledge boundary is inherently challenging, as there is no established formal definition. Our adaptation, determining the boundary by sampling multiple responses (inspired by [2]), is a practical choice that achieves a **good trade-off between reliability and computational cost**.
>     - **On label noise and the choice of $N$**: In our preliminary experiments, we varied the sampling number $N \in [8，16，32，64]$. As $N$ increases, the proportion of samples labeled as “OOB” (all incorrect among $N$ trials) decreases slightly and stabilizes as $N = 16$, suggesting that it offers a good balance between stability and cost. Additionally, the distribution of correct counts is bimodal, mainly concentrated at $N/N$ (clearly in-boundary) and $0/N$ (out-of-boundary). Samples with only a few correct trials (e.g., 1–2) are rare and have limited effect on boundary quality. We will include this preliminary experiment in the Appendix.
>     - **On computational overhead**: We agree that multi-sampling introduces additional cost, but this cost is **incurred only once** during data preprocessing stage for boundary estimation, will not incur during training or serving. This estimation, however, provides a more reliable reward environment for subsequent training, making the cost worthwhile.
>
>     [2] Reinforced Internal-External Knowledge Synergistic Reasoning for Efficient Adaptive Search Agent

---

> ### Author Response · Authors · 2025-11-22
> **Response to Questions (#1 - #3)**
>
> - Response to Question #1: CWR focuses on the fraction of wrong predictions made with high stated confidence. Our framework is explicitly designed to mitigate high-confidence errors by encouraging the model to recognize uncertain situations and avoid unwarranted confidence. In contrast, the Brier score and ECE are computed over all predictions and depend on how the full predicted probability distribution aligns with the observed labels, not only on high-confidence errors. Since we do not directly optimize these metrics and OOD accuracy remains roughly unchanged, the confidence reshaping that substantially reduces high-confidence wrong predictions can translate into only modest or mixed changes in Brier/ECE. This makes large gains in CWR compatible with mixed trends in Brier/ECE under OOD.
> - Response to Question #2: Thank you for this question, we will include the following clarifications in the revision.
>     - Ensemble size ($N$): As noted in our reply to Weakness #4, $N$ is selected based on preliminary experiments, balancing label stability and computational cost.
>     - Temperature and other parameters: For rollout and inference temperatures, as well as reinforcement learning hyperparameters, we performed standard grid search on a held-out validation set.
> - Response to Question #3: Thank you for pointing this out; we agree that this clarification improves the presentation.
>     - Model consistency: All Primus results are obtained from the official Primus checkpoints on HuggingFace, which are based on Llama-3.1-8B. We do not fine-tune Qwen-2.5 with Primus data. Following your suggestion, we additionally compare PrimusBase (the Llama-3.1 backbone used by Primus, pre-adapted with domain-specific security knowledge; the instruction-merged variant is not publicly available) against Qwen-2.5-7B (our backbone, without extra domain-specific pre-adaptation). The tables below show that PrimusBase is stronger on some benchmarks, but overall its performance is comparable to Qwen-2.5-7B. We will include this backbone comparison and details in the Appendix.
>
>
>         | Benchmarks | CTIBench |  |  |  |  | AttackSeqBench_Technique |  |  |  |  |
>         | --- | --- | --- | --- | --- | --- | --- | --- | --- | --- | --- |
>         | Metrics | Acc. | Brier | ECE | AUROC | CWR@0.8 | Acc. | Brier | ECE | AUROC | CWR@0.8 |
>         |  | (%)$\uparrow$ | $\downarrow$ | $\downarrow$ | $\uparrow$ | (%)$\downarrow$ | (%)$\uparrow$ | $\downarrow$ | $\downarrow$ | $\uparrow$ | (%)$\downarrow$ |
>         | PrimiusBase | 0.3679 | 0.5652 | 0.5639 | 0.4615 | 0.8209 | 0.275 | 0.6562 | 0.6575 | 0.3225 | 0.8276 |
>         | Qwen2.5-7B | 0.3491 | 0.5087 | 0.5387 | 0.6175 | 0.971 | 0.375 | 0.493 | 0.5225 | 0.68 | 1 |
>
>         | Benchmarks | CSEBench |  |  |  |  | CyberMetric |  |  |  |  |
>         | --- | --- | --- | --- | --- | --- | --- | --- | --- | --- | --- |
>         | Metrics | Acc. | Brier | ECE | AUROC | CWR@0.8 | Acc. | Brier | ECE | AUROC | CWR@0.8 |
>         |  | (%)$\uparrow$ | $\downarrow$ | $\downarrow$ | $\uparrow$ | (%)$\downarrow$ | (%)$\uparrow$ | $\downarrow$ | $\downarrow$ | $\uparrow$ | (%)$\downarrow$ |
>         | PrimiusBase | 0.364 | 0.6009 | 0.5918 | 0.3585 | 0.9057 | 0.3208 | 0.6412 | 0.6377 | 0.3727 | 0.9097 |
>         | Qwen2.5-7B | 0.4 | 0.4581 | 0.4687 | 0.5348 | 0.9931 | 0.4387 | 0.4526 | 0.466 | 0.6352 | 1 |
>
>         | Benchmarks | SEvenLLM |  |  |  |  | AttackSeq_Tactic |  |  |  |  |
>         | --- | --- | --- | --- | --- | --- | --- | --- | --- | --- | --- |
>         | Metrics | Acc. | Brier | ECE | AUROC | CWR@0.8 | Acc. | Brier | ECE | AUROC | CWR@0.8 |
>         |  | (%)$\uparrow$ | $\downarrow$ | $\downarrow$ | $\uparrow$ | (%)$\downarrow$ | (%)$\uparrow$ | $\downarrow$ | $\downarrow$ | $\uparrow$ | (%)$\downarrow$ |
>         | PrimiusBase | 0.3167 | 0.6205 | 0.6213 | 0.3562 | 0.8598 | 0.2162 | 0.7134 | 0.7214 | 0.3727 | 0.8707 |
>         | Qwen2.5-7B | 0.4392 | 0.4584 | 0.4642 | 0.559 | 1 | 0.428 | 0.4581 | 0.474 | 0.6418 | 1 |
>     - Overview of baseline variants: We will include detailed introductions of the baselines in the Appendix.
>         - Primus-Reasoning: a reasoning-oriented variant distilled from multi-step reasoning traces and reflection data generated by a larger reasoning LLM, specialized for cybersecurity reasoning tasks.
>         - Primus-Merged: a broader instruction-tuned variant that merges Primus-Instruct with Llama-3.1-8B-Instruct, improving general instruction-following ability while retaining reasoning skills.
>         - SecurityLLM (ZySec-7B): a domain-specific model fine-tuned on cybersecurity corpora, built upon HuggingFace’s Zephyr series, aiming to enhance knowledge and accuracy in security-related tasks.

---

> ### Author Response · Authors · 2025-11-22
> **Summary**
>
> Thank you again for the careful reading and constructive feedback, and for recognizing the strengths of our work. Our work aims to empowering LLMs with self-awareness for high-stakes tasks: we identify **self-awareness (knowledge-boundary and confidence-awareness)** as the missing capability behind overconfidence, and propose— to our knowledge, for the first time—an **RL framework that trains this self-awareness by decoupling awareness learning from task correctness**, then pairing it with adaptive inference strategies (e.g., RAG and low-confidence regeneration). This yields large, consistent reductions in confidently wrong outputs.
>
> In the above responses, we have addressed your concerns point-by-point, and we will add cross-domain evidence in the revision stage. If you have any further questions, please let us know—we would be happy to clarify. **If there are no remaining concerns, we kindly ask you to reconsider your evaluation in light of these clarifications.**

---

> > ### Comment · Reviewer_tSAQ · 2025-11-26
> >
> > Thanks for the detailed responses and the additional experiments, which cleared some of my concerns. I agree that empirically it seems to show little sign of hacking the CWR, but perhaps there could be a better design (or some additional metrics on which the method consistently beats baselines, given that metrics other than CWR do not improve in general). Overall, I think the authors should clarify the scope and contributions supported only by the empirical evidence presented, as other reviewers pointed out. I look forward to the cross-domain generalization results (W3) - I'll keep my score for now but happy to reassess accordingly.

---

> > > ### Author Response · Authors · 2025-12-04
> > > **Results from medical domain**
> > >
> > > Thank you very much for your **positive and constructive feedback**. We are glad that our previous response has addressed several of your concerns, and we will carefully incorporate your suggestions—especially regarding clarifying the scope and the contributions that are directly supported by the empirical evidence—into the revised version.
> > >
> > > In addition, the cross-domain generalization results (W3) on the **medical domain** are now available. We evaluate on the MedQA benchmark and report the following metrics:
> > >
> > > | Benchmarks       | MedQA    |          |          |          |             |
> > > | ---------------- | -------- | -------- | -------- | -------- | ----------- |
> > > | Metrics          | Acc. (%) | Brier    | ECE      | AUROC    | CWR@0.8 (%) |
> > > | Qwen2.5-7B       | 41.6     | 0.46     | **0.46** | 0.56     | 100         |
> > > | Ours (LLM)       | 40.9     | 0.47     | 0.47     | **0.60** | **77.3**    |
> > > | Ours (Framework) | **51.2** | **0.40** | 0.51     | 0.59     | 85.8        |
> > >
> > > From these results, we observe a substantial decrease in **CWR@0.8**, consistent with the findings in the cybersecurity domain. At the same time, the overall predictive performance of our framework remains comparable to the backbone Qwen2.5-7B (e.g., in terms of accuracy and other standard metrics), which is expected since we do not inject additional external knowledge, but instead focus on calibrating confidence and reducing overconfidence on high-risk instances.
> > >
> > > We hope these additional results further clarify the behavior and scope of our method, and we remain happy to reassess and refine the positioning of our contributions in line with the empirical evidence.

---

### Author Response · Authors · 2025-12-04
**Summary of the paper and review**

Dear Area Chair,

Thank you very much for your time and for helping handle this review cycle under such unusual constraints. We deeply appreciate your effort.
Below is a brief, factual summary intended to help contextualize the reviews of our submission.

------

## 1. Brief overview of the contribution

Our paper targets the problem of LLMs being overconfident errors in high-stakes settings (e.g., cybersecurity, medical decision support). We argue that overconfidence is not only a knowledge problem but also a capability problem: current LLMs lack self-awareness—the ability to recognize when a query is outside their knowledge boundary and to assess how confident they should be.

1. Formulating self-awareness as a first-class objective

   We decompose self-awareness into:

   - Boundary-awareness: judging whether a query is in- vs. out-of-boundary for the model’s knowledge.

   - Confidence-awareness: aligning verbalized confidence with the likelihood of correctness.

     We explicitly train these capabilities rather than relying on correctness-driven training or post-hoc calibration.

2. An RL framework decoupled from task correctness

   We train awareness signals via reinforcement learning with rewards not tied directly to correctness, encouraging the model to (i) recognize when correctness is unattainable and (ii) avoid being confident over wrong answers. Awareness learning is then paired with adaptive inference-time strategies such as retrieval-augmented generation and low-confidence regeneration.

3. Empirical validation in high-stakes domains

   Across cybersecurity and a medical domain, this substantially reduces confidently wrong outputs while maintaining competitive accuracy and calibration with the same backbone.

------

## 2. Summary of the review landscape

Across the four reviews:

- One “strong accept” (8, conf 4) with clear enthusiasm for the contribution.
- Two borderline-positive reviews (4, 4) — both state they would not oppose acceptance, and one notes our rebuttal plus new experiments significantly improved their view, and is happy to reassess according to the results from the medical domain.
- One negative review (0, conf 2), discussed below.

Most concerns raised by the first three reviewers (conceptual clarity, boundary definitions, metric choice, generalization, presentation issues) were either clarified or fully resolved during rebuttal. No reviewer besides zAPH raised the specific criticism detailed next.

------

## 3. Key factual clarification regarding zAPH’s review (0, conf 2)

We appreciate the strengths highlighted by zAPH. However, the core criticism relies on a factual misunderstanding of our method. The review assumes our framework requires internal confidence signals (logits, probabilities, gradient access) and that our RL objective operates on such internal uncertainties.
 In reality, our approach uses only verbalized confidence in natural language, explicit self-reported text outputs, and never accesses logits, hidden states, or model internals. Thus concerns about softmax temperature, logit magnitudes, or internal epistemic variables are not applicable.

To avoid further confusion:

- API-only models are inherently non-trainable, so the argument that our method requires “API-level applicability” is orthogonal; any RL/SFT method necessarily requires model access and this is not unique to our work.
- We do not aim to address open-world or inherently unknowable uncertainty; our focus is on high-stakes settings where confidently wrong outputs are unacceptable (cybersecurity, medical QA), not generic “real-world uncertainty.”
- Our improvement is not due to inference-time tricks: even without adaptive retrieval / regeneration, the awareness-trained model significantly outperforms strong baselines including XXX; the adaptive strategies simply operationalize the learned awareness at deployment.

Importantly, no other reviewer raised this misunderstanding, and the rest of the feedback, focused on clarity and scope, was addressed during rebuttal. We highlight this only so the review can be contextualized accurately, given frozen scores.

------

## 4. Overall strengths and limitations

Across reviewers there is clear agreement that:

- Addressing confidently-wrong behavior in high-stakes settings is important and timely;
- The conceptual framing and decoupling from correctness are novel and meaningful;
- The method is thoughtfully implemented and empirically solid within the domain;
- Results generalize across two high-stakes domains.

Limitations noted (imperfect calibration; not a generic calibration method) concern scope and positioning, and we have adjusted claims accordingly.

------

Given the above, we respectfully hope you will consider the submission for acceptance.
If any additional clarification would be helpful, we would be glad to provide it.

Thank you again for your time and for navigating this challenging cycle.

Best regards,
Authors of Submission 16164

---

> ### Author Response · Authors · 2025-12-04
> **Quick glance of the concerns raised by the reviewers and our responses**
>
> To help you quickly see the review landscape, we consolidated all weaknesses raised across reviewers, grouped them into thematic categories, and indicated whether they have been resolved, clarified in the responses or is a misunderstanding . This table summarizes the overall picture clearly:
>
> | Weakness Category                                            | Mentioned by                        | Resolved?                      | Notes                                                        |
> | ------------------------------------------------------------ | ----------------------------------- | ------------------------------ | ------------------------------------------------------------ |
> | (1) Conceptual clarity; distinguishes with existing work.    | tSAQ (w1), LaYM (w1, w2), NfsF (w3) | Clarified and resolved         | Will revise for clarity, added formal definitions.           |
> | (2) Boundary-awareness definition; IB/OOB labeling validity  | tSAQ (w4), LaYM (w6)                | Clarified and resolved         | Added rationale for the reason underlying the choices        |
> | (3) Concerns about the CWR metric                            | tSAQ (w3), LaYM (w4)                | Clarified                      | Provided rationale for 0.8+ region; showed consistent trends across thresholds. |
> | (4) Underperformance on standard metrics (Accuracy, ECE, AUROC) comparing with knowledge-injection baselines | zAPH (w4), LaYM (w3), NfsF (w1)     | Clarified                      | Explained focus on reducing confidently wrong errors; baselines have heavy domain-specific supervision. |
> | (5)  Minor clarity / presentation issues / related work      | NfsF (w1, w2, w4)                   | Resolved                       | Will revise for clarity (abstract phrasing, figure references, terminology) and add related work |
> | (6) Generalization in other high-stakes domain               | tSAQ (w2), NfsF(w5), zAPH(w3)       | Resolved                       | Added medical-domain results; clarified intended scope as high-stakes tasks. |
> | (7) Missing calibration baselines                            | LaYM (w5)                           | Clarified and Acknowledged     | Clarified we are not proposing a generic calibration method  |
> | (8) Assumes logit-based internal confidence & gradient access,  applicability to closed-weight models (API-only) | zAPH (w1, w2)                       | Misunderstanding and Clarified | Our method uses *only verbalized confidence* (text outputs), no logits or internal signals; explained in rebuttal. Any RL/SFT approach requires model access; this is not unique to our method. |

---

### Meta-Review · Area_Chair_P2Ym · 2026-01-02

**Summary:**

This paper proposes a reinforcement-learning framework to equip large language models with “self-awareness” for high-stakes tasks such as cybersecurity analysis and medical decision support. The motivation is that LLMs are widely known to exhibit overconfident behavior, which can lead to serious consequences in high-stakes settings. The authors define self-awareness as consisting of two components: boundary-awareness, i.e., knowing whether a query falls within the model’s knowledge scope, and confidence-awareness, i.e., assessing how likely the model’s response is to be correct. Corresponding reward functions are designed for each component, and the model is trained using GRPO. The method is evaluated primarily on cybersecurity benchmarks, where it is shown to reduce failures in tail-risk cases (making errors with high confidence).

**Reviewer Concerns:**

Reviewer tSAQ
- Clarifying contributions relative to prior work: Resolved. The rebuttal clarifies the distinction from prior work (e.g., RLCR), particularly in terms of decoupling awareness learning from correctness and introducing an explicit boundary-awareness objective.
- Generalization beyond cybersecurity: Resolved. The authors provided additional experimental results in the medical domain, addressing the reviewer’s concern about domain specificity.
- Strength of empirical results: Not resolved. While the authors reframed the objective toward reducing high-confidence errors, the concern that results are not strong on standard accuracy and calibration metrics remains.
- Potential flaws or “gaming” of the CWR metric: Resolved. The authors’ rebuttal and additional analysis reasonably address the concern that CWR improvements arise from trivial confidence suppression.
- Simplified in-boundary vs. out-of-boundary labeling heuristic: Resolved. The authors provided additional justification and sensitivity analysis, explaining this choice as a practical compromise with limited impact on the main conclusions.

Reviewer zAPH
- Limited applicability due to “white-box” requirements (logits/weights access): Resolved (misunderstanding). The rebuttal clarifies that the method does not rely on logits or internal confidence signals; it operates on verbalized confidence in generated text. While model-weight access is required for RL fine-tuning, this is a general constraint shared by essentially all tuning-based methods and is not specific to this approach.
- Lack of theoretical connection between logits and confidence: Resolved (misunderstanding). This critique is largely inapplicable given that the approach does not interpret logit magnitude as uncertainty; it evaluates and trains text-level self-reported confidence behaviorally.
- Evaluation design does not reflect natural failure modes: Partially addressed / not fully resolved. The authors respond that the evaluation is intentionally scoped to high-stakes domains with operationally defined boundaries.
- No disentanglement between inference-time adaptive strategies and self-awareness rewards: Partially addressed. The rebuttal points to ablations and argues that awareness training alone already reduces high-confidence errors, while adaptive strategies mainly affect task completion/accuracy.

Reviewer NfsF
- Overselling the significance of the results: Not resolved. The reviewer’s remains valid. While the authors acknowledge this point and indicate they would tone down claims, doing so would require a substantial reframing of the paper.
- Generalization to domains without a single ground-truth answer: Not resolved, but largely out of scope.

Reviewer LaYM
- Lack of clear definitions for newly introduced terms: Resolved. The authors provided clearer definitions in the rebuttal and committed to incorporating them in the revision.
- Ambiguity between boundary-awareness and confidence-awareness: Partially resolved. The authors acknowledge that the separation may not be strictly necessary from a theoretical standpoint, but argue that it is practically useful for deployment-oriented decision pipelines.
- Empirical performance relative to baselines: Not resolved. The concern that the proposed method frequently underperforms strong baselines in terms of accuracy and standard uncertainty/calibration metrics remains. This is a recurring issue raised by multiple reviewers.
 Sensitivity of CWR to threshold choice: Resolved. The authors provide additional analysis showing consistent improvements in CWR across multiple confidence thresholds.
- Missing calibration baselines: Not resolved. The authors argue that their method is not intended as a generic calibration approach, but rather as a framework for reducing high-stakes failures. However, given that confidence-awareness is a central component of the method, the absence of strong calibration or hallucination-detection baselines remains a valid concern.
- Validity of the IB/OOB distinction: Partially resolved. The authors acknowledge the heuristic nature of the boundary definition and provide evidence of robustness to sampling size and prompt choice.

**Reviewer Scores:**

After reviewing all referee reports and the corresponding rebuttals, I conclude that none of the initially negative reviewers would ultimately champion the paper. I am largely aligned with Reviewer LaYM. Multiple reviewers consistently point out that the proposed method underperforms strong baselines on standard accuracy and uncertainty quantification metrics. While the authors argue that their method is not intended to directly improve these metrics and instead targets tail-risk failures measured by CWR, standard calibration metrics such as Brier score and ECE already penalize high-confidence errors and are therefore directly relevant to safety-critical evaluation. As such, they cannot be treated as orthogonal to tail-risk behavior.

At its current stage, the paper relies too heavily on CWR to justify its contribution and would require substantial reframing—both conceptually and empirically—to clearly position tail-risk reduction as its primary objective and to support this framing with an appropriate evaluation protocol. These changes go beyond what can reasonably be addressed in a camera-ready revision.

---

### Decision · Program_Chairs · 2026-01-26

Reject